# B chromosome retrotransposed sequences persist through speciation, contributing to genomic and regulatory innovations in the fish genus *Psalidodon* (Characiformes, Acestrorhamphidae)

Lucas F. Lasmar[1], Mateus R. Vidal[1], Pamela C. F. Nadai[1], Rodrigo Zeni dos Santos[2], Raquel da Costa Machado[1], Ricardo Utsunomia[2], Francisco J. Ruiz-Ruano[3,4], Alexander Suh[3,4,5], Claudio Oliveira[1], Duilio M. Z. A. Silva[1]*, Fausto Foresti[1]

**1** Departamento de Biologia Estrutural e Funcional, Instituto de Biociências de Botucatu, Universidade Estadual Paulista, UNESP, Botucatu, São Paulo, Brazil, **2** Departamento de Ciências Biológicas, Faculdade de Ciências, Universidade Estadual Paulista, UNESP, Campus de Bauru, Bauru, São Paulo, Brazil, **3** Centre for Molecular Biodiversity Research, Leibniz Institute for the Analysis of Biodiversity Change, Bonn, North Rhine-Westphalia, Germany, **4** Bonn Institute for Organismal Biology (BIOB) – Animal Biodiversity, University of Bonn, Bonn, North Rhine-Westphalia, Germany, **5** Department of Organismal Biology – Systematic Biology, Evolutionary Biology Centre, Science for Life Laboratory, Uppsala University, Uppsala, Sweden

* duilio.silva@unesp.br

## Abstract

B chromosomes are supernumerary genetic elements rich in repetitive DNA. Many species within the fish genus *Psalidodon* possess a large metacentric B chromosome that exhibits signs of recent retrotransposon activity, resulting in truncated pseudogenic copies of standard A chromosomes genes, specifically *sbno2* and *simc1*. We aimed to characterize the structure of the B chromosome pseudogenes *sbno2-B* and *simc1-B* and their evolutionary history in four B chromosome variants of three *Psalidodon* species, as well as their expression patterns in *Psalidodon paranae* individuals with a single B chromosome (1B). Our findings suggest that the retrotransposition process of each gene occurred in an ancestral B chromosome, which later diverged into distinct trajectories within each species following speciation. In the high-copy B chromosomes, the *sbno2-B* pseudogene shows dozens of interspersed copies, while the *simc1-B* pseudogene usually shows three copies arranged in tandem. In *Psalidodon paranae* 1B individuals, underexpression of the *sbno2* and *simc1* genes in ovaries indicates that the pseudogenic B copies probably influence gene expression through non-coding RNA interference mechanisms. This shows that the retrotransposon activity on B chromosomes generates genomic diversity and neofunctionalization, impacting gene regulatory networks, and possibly contributing to the persistence of B chromosome in populations.

**Data availability statement:** All sequencing files are available from the GenBank database (accession numbers SAMN47549717, SAMN47549721, SAMN47549722, SAMN47549718, SAMN47549719, SAMN47549720, SAMN47549711, SAMN47549714, SAMN47549716, SAMN47549712, SAMN47549713, SAMN47549715, SAMN47549724, SAMN47549726, SAMN47549727, SAMN47549723, SAMN47549725, SAMN47549728.).

**Funding:** This study received financial support from the Conselho Nacional de Desenvolvimento Científico e Tecnológico (CNPq) https://www.gov.br/cnpq/pt-br and the Coordenação de Aperfeiçoamento de Pessoal de Nível Superior (Capes) https://www.gov.br/capes/pt-br. The authors D. M. Z. A. S. and F. F. received grants from the Fundação de Amparo à Pesquisa do Estado de São Paulo (grants 2017/22447-8, 2020/01775-0) https://fapesp.br/. The authors F. J. R. R. and A. S. received grants from the European Research Council (ERC), European Union (grants 0000-0002-5391-301X, 101002158 GermlineChrom) https://erc.europa.eu/homepage. The author A. S. received a grant from the Swedish Research Council Vetenskapsrådet (grant 2020-04436) https://www.vr.se/english.html. Funders did not play any role in the study design, data collection and analysis, decision to publish, or preparation of the manuscript.

**Competing interests:** The authors have declared that no competing interests exist.

## Introduction

B chromosomes and retrotransposons are distinct genetic elements that share a remarkable ability to selfishly take advantage of biological processes, thereby increasing their transmission to future generations through different molecular mechanisms [1,2]. B chromosomes are supernumerary genetic elements that primarily cheat by preferentially being incorporated into generative cells during meiosis [2]. However, in some species, B chromosomes are also known to exhibit drive (biased transmission) mechanisms at pre-meiotic and post-meiotic stages [2]. In contrast, retrotransposons are genomic sequences that over-replicate during cell cycles [1]. Retrotransposons jump to new genomic locations through a copy-and-paste mechanism, utilizing the cell's transcriptional machinery [3], and some retrotransposons can accidentally retrotranspose mRNAs of host genes [4]. Because retrotransposition involves an mRNA intermediate, retrotransposed genes typically lack introns and regulatory elements [5]. A key feature of the retrotransposed sequences is the presence of exon-exon junctions in the genomic DNA (gDNA) copies. Intriguingly, retrotransposons can hitch a ride on B chromosomes, contributing to the latter's increased accumulation in populations [6–8].

With the increasing accessibility of technologies for high-throughput sequencing methods, pseudogenes have been identified on B chromosomes of several eukaryote species [9,10]. On the other hand, few studies have shown the presence of retrotransposed pseudogenes on the B chromosome. The only research conducted on this topic thus far is by Carmello et al. [11], who demonstrated in fishes, the *hnRNP Q-like* pseudogene was retrotransposed onto the B chromosome of *Astatotilapia latifasciata*.

The genus *Psalidodon* Eigenmann, 1911 within the order Characiformes includes about 30 species, at least eight of which have B chromosomes [12]. These B chromosomes are highly morphologically diverse from micro to macrochromosomes with varying shapes. The most common variant in the genus is the large metacentric B chromosome [12]. The metacentric B chromosome in at least four *Psalidodon* species is believed to have originated from a common ancestor approximately 4 million years ago [13]. These B chromosomes share several genes related to their evolutionary success, including those involved in cell-cycle regulation, chromosome segregation, and sexual development [13]. Additionally, they also contain truncated pseudogenic copies of functional genes, such as the *amhr2* [13]. A deeper look into the data of Silva et al. [13], revealed that short-read sequences from B chromosome-carrying samples (1B) in *Psalidodon paranae*, mapped to exon-exon junctions of the genes *sbno2* and *simc1,* suggesting the presence of retrotransposed pseudogenes (*sbno2-B* and *simc1-B*) on the B chromosome.

The goal of this study was to investigate the evolutionary history of these two pseudogenes, *sbno2-B* and *simc1-B,* on the B chromosomes in *Psalidodon* species, as well as their potential effects on gene expression. Our results show that the *sbno2-B* and *simc1-B* pseudogenes were retrotransposed onto an ancestral *Psalidodon* B chromosome. These pseudogenic B copies survived through speciation events and followed independent evolutionary paths in each species. Furthermore,

we observed that the functional canonical *sbno2* and *simc1* genes are underexpressed in 1B individuals of *P. paranae,* suggesting that the pseudogenic copies on the B chromosome influence the expression of the canonical genes of the A chromosomes, possibly interfering with molecular pathways for their own benefit.

## Materials and methods

### Sampling and cytogenetic analyses

We analyzed samples of *P. paranae*, *P. bockmanni,* and *P. fasciatus* from four populations with B chromosomes from São Paulo State, Brazil. The samples, localities, and B chromosome types are described in the S1 Table in S1 Table. After analysis, we deposited the specimens in the fish collection of the Fish Biology and Genetics Laboratory (LBGP) at UNESP, Botucatu, São Paulo, Brazil under vouchers described in S1 Table in S1 Table.

The animals were collected in accordance with the precepts of Law 11,794 of October 8, 2008, with Decree 6,899 of July 15, 2009, as well as guidelines set forth by the National Council for the Control of Animal Experimentation (CONCEA), and was approved by the Ethics Committee on the Use of Animals of the São Paulo State University/Institute of Biosciences (IBB/UNESP) at the meeting on 02/09/2022.

Mitotic chromosomes were obtained following the protocol established by Foresti et al. [14]. The chromosome morphology was determined based on the arm ratio method described by Levan et al. [15], classifying the chromosomes as metacentric (m), submetacentric (sm), subtelocentric (st), and acrocentric (a). To detect the partially heterochromatic B chromosomes in the individuals of each population, we used the C-banding technique originally described by Sumner [16], with some adaptations.

### Genomic DNA extraction

We extracted genomic DNA from muscle and fin clips. For muscle, we used the NucleoSpin® Tissue Columns Kit with residual RNA removal by RNase A (20 mg/ml – Invitrogen). For fin clips, we used the "Hot Shot" protocol proposed by Meeker et al. [17]. The samples are described in the S2 Table in S1 Table.

### Sequencing and coverage analysis

Genomic DNA (gDNA) short reads were obtained from 0B (animals without B chromosome) and 1B samples of *P. bockmanni* and specimens from two populations of *P. fasciatus* using the BGISEQ-500 platform (BGI Shenzhen Corporation, Shenzhen, China). The sequencing information and used libraries are summarized in S3 Table in S1 Table. For *P. paranae*, we used the gDNA short reads obtained by Silva et al. [13]. Long non-coding RNAseq reads of *P. paranae* were obtained from dos Santos et al. [18].

For the DNA and RNA coverage analysis, the reads were first checked for quality and adapters were removed with Trimmomatic [19], then the mapping of the *sbno2* and *simc1* sequences was carried out as described in Silva et al. [13] with the SSAHA2 software [20] and a minimum alignment score of 40 and 80% of minimum identity. Finally, we used customized scripts to count the number of reads mapped as a measure of abundance and to generate the coverage graphics (https://github.com/fjruizruano/ngs-protocols/blob/master/mapping_blat_gs.py; https://github.com/fjruizruano/ngs-protocols/blob/master/coverage_graphics.py). RNA coverage analysis was performed only for *P. paranae* using previously obtained RNAseq libraries from Silva et al. [13].

### Search for in-tandem repeated sequences and mobile elements

Due to the *sbno2-B* and *simc1-B* copy number amplification on the B chromosomes, we checked if those sequences were repeated in tandem. For this, we first selected pairs of reads showing similarity with the *sbno2-B* and *simc1-B* sequences (12853 pairs for *sbno2* and 12552 pairs for *simc1*). Then, the selected reads were input into TAREAN [21] using standard

parameters, for k-mer decomposition and reconstruction of possible tandemly repeated sequences. Additionally, we checked whether the *sbno2-B* and *simc1-B* sequences have similarities with the *P. paranae* satellite DNAs described in the literature [22] by running a homology search using RepeatMasker (https://github.com/fjruizruano/ngs-protocols/blob/master/rm_homology_v2.py) and a *de novo* assemble using Geneious Prime 2024.0.7.

Finally, to check the pseudogenes structure in long reads, we analyze PacBio long reads from a 1B sample [18] (SRA accession number: SRR11678219) similar to the *sbno2-B* and *simc1-B* sequences by a BLASTn search with e-value of 1e-10 and minimum query cover of 70%. Then, we selected the regions from each read including the pseudogene and 10 kb from each flank for annotation using Geneious. To check if the pseudogene sequences occupy different locations on the B chromosome, we selected 1000 bp of each flank of the pseudogenes on each PacBio read and performed a high sensitivity *de novo* assembly using Geneious software. If the flanking sequences were similar, we assumed the PacBio reads came from the same genome location. If they were distinct, we assumed they came from different locations of the B chromosome. The flanking sequences (4000 bp on each side) of the PacBio reads recovered for the *simc1-B* pseudogene were screened for repeats using CENSOR [23]. We did not screen the *sbno2* PacBio reads due to the high number of reads obtained that would not be informative.

**Quantitative Reverse Transcription PCR (RT-qPCR)**

To confirm the RNA coverage analysis, we measured the expression levels of the *sbno2* and *simc1* genes by RT-qPCR. Quantification of expression levels of *sbno2* and *simc1* genes was performed using ovaries from 0B and 1B females with three biological replicates per group. Tissue manipulation, RNA extraction and RT-qPCR reactions were performed as in Silva et al. [13,24] using the primers described in the S4 Table in S1 Table. The target and reference genes were simultaneously analyzed in duplicate across two independent samples. The normalized relative expression quantity (NREQ) was calculated using the $2^{-\Delta\Delta Cq}$ method [25], with hypoxanthine phosphoribosyltransferase 1 (*hprt1*) serving as the reference gene, and results were subsequently calibrated to the mean expression level of the 0B group. Two-group comparisons were performed by the Gardner-Altman estimation plot method devised by Ho et al. [26] following Gardner and Altman's design [27], as implemented on https://www.estimationstats.com/.

**Exon-exon junction analysis**

To check for the presence of exon-exon junctions in *sbno2* and *simc1* sequences we extracted the mapped reads in the previous coverage analysis using a custom script (https://github.com/fjruizruano/ngs-protocols/blob/master/mapping_blat_gs.py). The selected reads were then mapped against the sequence of each gene with the Geneious® 7.1.3 software using the Geneious mapper algorithm. Then, we manually checked the results for the presence of reads mapping at both sides of the exon-exon junctions. To validate this analysis, we designed primers for use in multiplex PCR reactions using a modified version of Primer3 2.3.7 [28] implemented in the Geneious® 7.1.3 software. The sets of primers for both the *sbno2* and *simc1* genes contain a forward primer within an exon and two reverse primers, one within an exon and another one in the exon-exon junctions (S1 Fig) (S4 Table in S1 Table). With this approach, we expect a short single amplicon in 0B samples from the A copies and two amplicons in the 1B, the short one from the A and B copies, and another longer one exclusive of the B copies. The primer pairs sets were designed to generate 84 bp and 239 bp amplicons for *sbno2* and 80 bp and 319 bp amplicons for *simc1*. We also developed a set of primer pairs for the exon-exon junction 9–10 of the *sbno2-B* pseudogene for use in a regular PCR to confirm the presence of this junction in the 1B and 0B samples (S4 Table in S1 Table, S1 Fig). The expected amplicon size for this 9–10 exon junction was 76 bp (S1 Fig). PCR reactions were carried out using 15 ng of DNA, 1X PCR buffer, 1.5 mM MgCl$_2$, 200 μM dNTPs, 0.1 μM of each primer and 1U Taq DNA polymerase in a final volume of 12.5 μl. Thermocycling conditions were initial denaturation at 94°C for 2 min, 35 cycles at 94°C for 45 s, 58°C for 45 s, 68°C for 30 s and a final extension step at 68°C for 10 min. Only for *P. fasciatus* (BfMa) from the Piratininga stream, the annealing temperature used was 66°C.

                                                                                                

### Search for B-specific Single Nucleotide Polymorphisms (SNPs) in *Psalidodon* species

The search for SNPs in the pseudogenes of the B chromosomes was performed using customized scripts (https://github.com/fjruizruano/whatGene and https://github.com/fjruizruano/ngs-protocols). For *P. paranae* we used four gDNA and six RNA libraries from Silva et al. [13] and six lncRNA libraries previously obtained from dos Santos et al. [18]. Additionally, we used six libraries for both *P. fasciatus* and *P. bockmanni* (12 in total) obtained in the present study (S3 Table in S1 Table). We mapped each library against *sbno2* and *simc1* sequences using "mapping_blat_gs.py" script (https://github.com/fjruiz-ruano/ngs-protocols/blob/master/mapping_blat_gs.py), with SSAHA2 option (v.2.5.5) [20]. Then, using SAMtools (v.1.3.1) [29] we SNP calling to obtain putative SNPs exclusive from the 1B libraries. After putative merged the bam files in each respective group, i.e., 0B gDNA, 1B gDNA, 0B RNA, 1B RNA, 0B lncRNA, and 1B lncRNA; and separately for each species. After merging, we used the "bam_var_join.py" script (https://github.com/fjruizruano/whatGene/blob/master/scripts/bam_var_join.py) to obtain the counts per position of each nucleotide, and insertions/deletions. Then, with the "snp_calling_bchr.py" script (https://github.com/fjruizruano/whatGene/blob/master/scripts/snp_calling_bchr.py) we performed SNP selection for each group, we extracted counts of the Ref (reference) and Alt (alternative) alleles for each library. Then, for each type of library (gDNA, RNA and lncRNA) we used two filters in the putative SNPs. In the first filter, we only kept the SNPs with the occurrence of Alt alleles in all 1B individuals of the group. For the second filter, we obtained the mean value of occurrence in each position for the 0B libraries. Considering the presence of two copies of each gene on the A chromosomes, and at least one copy on the B, we discarded all the Alt alleles with occurrence of less than half of the 0B mean.

### Fluorescence in situ Hybridization (FISH)

The FISH probe for the *sbno2* gene was generated by PCR including digoxigenin-11-dUTP (Roche Applied Science) in the reaction with the primers designed by Silva et al. [13] (S4 Table in S1 Table). FISH experiments were carried out as in Pinkel et al. [30] pin, using high stringency conditions, and the signals were detected with anti-digoxigenin-rhodamine (Roche Applied Science). Chromosomes were counterstained with 4',6-Diamidino-2-phenylindole (DAPI) (Vector Laboratories, Burlingame, CA). Images were captured with a digital camera (Olympus DP90) attached to an Olympus BX61 epifluorescence photomicroscope and acquired using CellSens Dimension (Olympus). Image treatment, optimization of brightness, and contrast were performed using the Adobe Photoshop CS4 program. Our investigation approach is summarized in S2 Fig.

## Results

When mapping short-read sequence data against the A chromosomes, the *sbno2* and *simc1* genes show regions with higher gDNA coverage in 1B samples across all B chromosome variants analyzed, except for the *sbno2* gene on the BbM variant (metacentric B chromosome of *P. bockmanni*) and the *simc1* gene on the BfMa (metacentric B chromosome variant "a" of *P. fasciatus*) (Fig 1a, S3 Fig). These high-coverage regions span portions of each of the two genes, suggesting these B-variants have truncated pseudogenic copies, referred to as *sbno2-B* and *simc1-B*, of the canonical genes on the A chromosomes. The high coverage regions among species are coincidental for each gene, i.e., the copies on the B chromosomes are truncated in the same regions for each gene (Fig 1a). Furthermore, the coverage graphs show that different B chromosome variants have distinct copy numbers of the pseudogenes (Fig 1a), ranging from eight on the BfM to 125 on the BpM for *sbno2-B*, and from two on the BfMb and BbM to three on the BpM for *simc1-B* (Fig 1a).

We also analyzed *Psalidodon* B chromosomes for complete copies (with introns) and non-truncated retrogene copies (without introns) of the *sbno2* and *simc1* genes. To do this, we mapped gDNA reads of 0B and 1B samples of the four *Psalidodon* populations analyzed here onto the *sbno2* and *simc1* full gene sequences, including introns. This mapping revealed no coverage differences in intronic regions between 0B and 1B groups across any species (S4a, b Fig), except for small areas composed of fragments of genes (S4c, d Fig, S5 Table in S1 Table). These results suggest that the *Psalidodon* B chromosomes only contain retrotransposed truncated copies of the s*bno2* and *simc1* genes, with no full gene sequences.

 

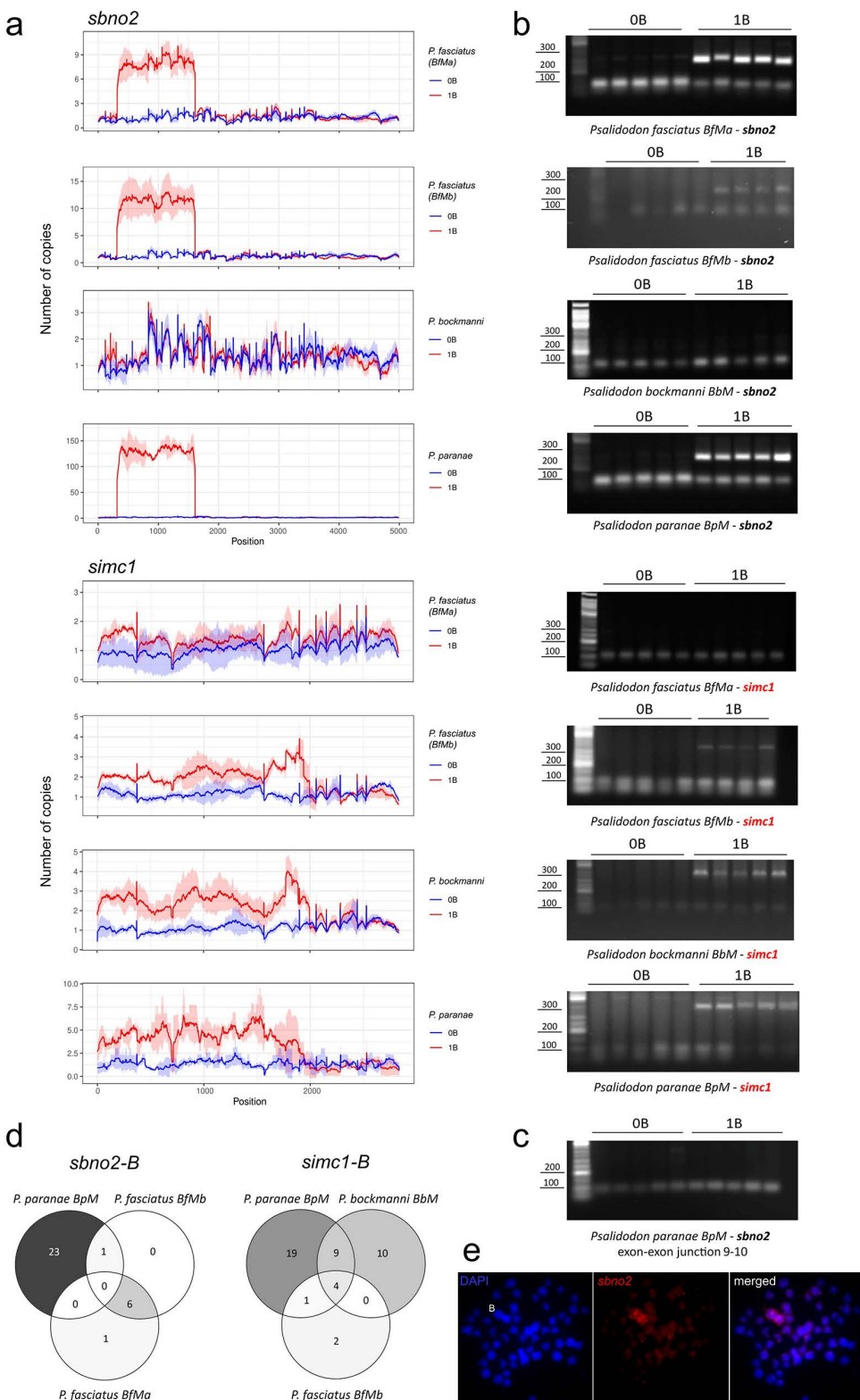

**Fig 1. Detection of pseudogenes on the B chromosomes of *Psalidodon* species. a** coverage graphs of the pseudogenes on the B chromosomes of *Psalidodon* species (the *P. paranae* graph was adapted from Silva et al., 2021). **b** electrophoresis agarose gels showing the amplification of B-specific

pseudogenic sequences (upper band) of the genes *sbno2* and *simc1* on samples with B chromosomes (1B) by multiplex PCR. The lower band on all samples is a positive control (see methods section). **c** amplification of the exon-exon junction 9-10 in both 0B and 1B samples by regular PCR. **d** Venn diagram showing the number of B-specific SNPs shared among the B chromosome variants of the *Psalidodon* species analyzed in this study. **e** Metaphase of *P. paranae* after FISH with a probe for the *sbno2* gene. Bar = 10 μm.

TAREAN clustering analysis did not detect the typical satellite DNA clusters for the *sbno2* and *simc1* sequences, indicating that the amplified copies of these genes on the B chromosome of *P. paranae* are not tandemly arranged (Supplementary Material S1, S2 Files). PacBio long read analysis confirmed this for the *sbno2-B* pseudogene (S5a Fig). However, for the *simc1-B* pseudogene, PacBio reads revealed an in-tandem structure consisting of three gene copies, consistent with the copy number estimated by short reads coverage (S5b Fig, Fig 1a). Comparison of the *sbno2-B* and *simc1-B* sequences to the *P. paranae* satellitome (Silva et al., 2017) [22] revealed no significant sequence similarities.

To determine whether the pseudogene sequences are located in different regions of the B chromosome, we recovered 169 PacBio long reads encompassing the *sbno2-B* pseudogene. Analysis of the gene flanks (1000 bp on each side) revealed more than 70 distinct sequences. Additionally, about 70 sequences assembled into a single contig, while other small groups formed separate contigs. This suggests that the *sbno2-B* sequences originate from different paralogs from distinct regions of the B chromosome. FISH mapping further supported this, confirming the presence of the *sbno2* gene at several locations on the *P. paranae* B chromosome (Fig 1e), with prominent clusters symmetrically positioned on the arms and smaller clusters scattered throughout. For the *simc1-B* pseudogene, only five PacBio reads were recovered which assembled into a single contig with three paralog copies (S5b Fig) indicating that they derive from the same genomic sequence. FISH did not show visible signals for the *simc1* gene, likely due to its low copy number on the B chromosome. Repeat screening of the *simc1-B* and *sbno2-B* flanking sequences did not identify the presence of repetitive elements in or near the pseudogenes with high confidence.

gDNA read mapping also revealed reads spanning the exon-exon junctions in high-coverage regions of the *sbno2-B* and *simc1-B* pseudogenes in 1B samples (S6a, c Fig) as detailed in Table 1, confirming the absence of introns on the B chromosome copies of these genes. Multiplex PCR reactions further confirmed the presence of these exon-exon junctions in the 1B samples (Table 1, Fig 1b). Additionally, gDNA reads spanning exon-exon junctions 8–9 and 9–10 were detected in a 0B individual of *P. paranae* (S6b Fig). Although these reads were present in only one 0B individual, PCR assays with five 0B individuals confirmed that junction 9–10 was present in all samples (Fig 1d). These PCR results, obtained from different individuals than those used for Illumina sequencing, rule out sequencing errors such as index swapping [31], confirming that in 0B samples, a region containing exons 8–10 without intervening introns was also retrotransposed to some part of genome A.

**Table 1. Summary of read mapping and multiplex PCR results for exon-exon junctions in the high-coverage B-regions of the *sbno2* and *simc1* genes in 1B and 0B samples.**

| Species | B-variant | *sbno2* 1B | *sbno2* 0B | *simc1* 1B | *simc1* 0B |
|---|---|---|---|---|---|
| *P. paranae* | BpM | all (2) | 8-9, 9-10(1*) | all (2) | none (1) |
| *P. bockmanni* | BbM | none (1) | none (1) | all (2) | none (1) |
| *P. fasciatus* | BfMa | all (2) | none (1) | none (1) | none (1) |
| | BfMb | all (2) | none (1) | all (2) | none (1) |

"all" indicates the presence of reads across all exon-exon junctions within the high-coverage B-region, and "none" indicates the absence of reads in all exon-exon junctions. The numbers 8–9 and 9–10 refer to the specific exon-exon junctions where reads were detected. (1) indicates one amplicon obtained in the multiplex PCR, (2) indicates two amplicons were obtained, and (1)* indicates one band was amplified in the regular PCR. For further details, refer to the methods section.

If the 0B samples have more copies of the fragments including exons 8–10, we would expect higher coverage in this region relative to its flanking regions in the coverage graphs. As expected, we observed a slightly higher coverage of this region compared to its flanks (S7 Fig).

The number of B-specific SNPs for the *sbno2-B* and *simc1-B* pseudogenes in gDNA in the high-coverage regions for each species is summarized in the Venn diagrams in Fig 1d (S8-S15 Tables in S1 Table). For both the *sbno2-B* and *simc1-B* pseudogenes, *P. paranae* exhibited the highest number of SNPs (24 and 33, respectively). For the *sbno2-B* pseudogene, *P. fasciatus* (BfMa) and *P. fasciatus* (BfMb) shared the highest number of B-specific SNPs (7 each), with no B-specific SNP common to all species. For the *simc1-B* pseudogene, *P. paranae* and *P. bockmanni* shared the highest number of B-specific SNPs (13), with four of these also present in *P. fasciatus* (BfMb). Both populations of *P. fasciatus* had the lowest number of B-specific SNPs for both pseudogenes.

The RNA coverage analysis for mRNA and lncRNA reads for the *sbno2* and *simc1* genes in 0B and 1B ovaries of *P. paranae* showed that both genes are underexpressed in 1B samples (Fig 2a). In contrast, RNA coverage in 0B samples was uniform throughout the coding sequences for both genes. RT-qPCR for mRNA confirmed the expression patterns for both genes (Fig 2b, S6, S7 Tables in S1 Table) with high 0B/1B fold change values, FC = 25.4 and 18.2 for the *sbno2* and *simc1* genes, respectively. Because of the low number of B-specific SNPs, expression levels were measured for all copies (A+B) per gene together.

The SNP analysis using lncRNA reads did not show B-specific SNPs for the *sbno2* and *simc1* genes, indicating that the B chromosome does not express long non-coding sequences with B-specific variation (S16, 17 tables in S1 Table). We analyzed the mRNA reads and detected three SNPs for the *sbno2* gene in the high-coverage region and zero SNPs for the *simc1* gene. This strongly indicates that mRNA transcripts of the *sbno2* gene from the B chromosome are expressed (S18, 19 tables in S1 Table). Considering that our filters were very stringent for this analysis (see methods section), we cannot rule out that copies of the *simc1* gene are expressed from the B chromosome, since one individual exhibited SNPs on the high-coverage region of this gene, but in a low proportion compared to the 0B reference copies (S19 Table in S1 Table).

We created a quick protocol for identifying individuals bearing B chromosomes extracting DNA with the HotShot protocol and doing multiplex PCRs with A-specific and B-specific primers combined (full protocol detailed in the Methods section). Using the primers set for the *sbno2* or *simc1* genes, the protocol can be applied to all the B-variants analyzed here. Using this approach, the genotyping time was 3 hours, from tissue collection to results.

## Discussion

Here we demonstrate that four B chromosome variants of three *Psalidodon* species have retrotransposed copies of the *sbno2* and *simc1* genes that were acquired in a common ancestor and maintained across speciation events. These findings were based on the presence of exon-exon junctions for the B copies of these genes at DNA level, confirmed by bioinformatic analyses of gDNA reads and multiplex PCR. Furthermore, the coverage analyses for the *sbno2* and *simc1* genes showed that the pseudogenic B copies of each gene are truncated in the same regions on the B-variants analyzed here. This similarity indicates that the *sbno2-B* and *simc1-B* pseudogenes originated on an ancestral *Psalidodon* B chromosome through retrotransposition events from mRNAs of the copies present on the standard A chromosomes, as illustrated in Fig 3. This evidence of retrotransposition of gene fragments to the B chromosome is reinforced by the absence of introns for the *sbno2-B* and *simc1-B* pseudogenes on the *Psalidodon* B chromosomes (see results). This allows us to discard the possibility of an initial origin or colonization of the B chromosome with full copies of the genes followed by internal retrotranspositions within the B chromosome.

The metacentric B chromosomes of the genus *Psalidodon* share a common ancestor origin at least 4 million years ago [13]. The number of shared SNPs among each pair of *Psalidodon* B chromosomes here is correlated with the phylogenetic history of the genus [32], which shows *P. paranae* and *P. bockmanni* phylogenetically closer than these species

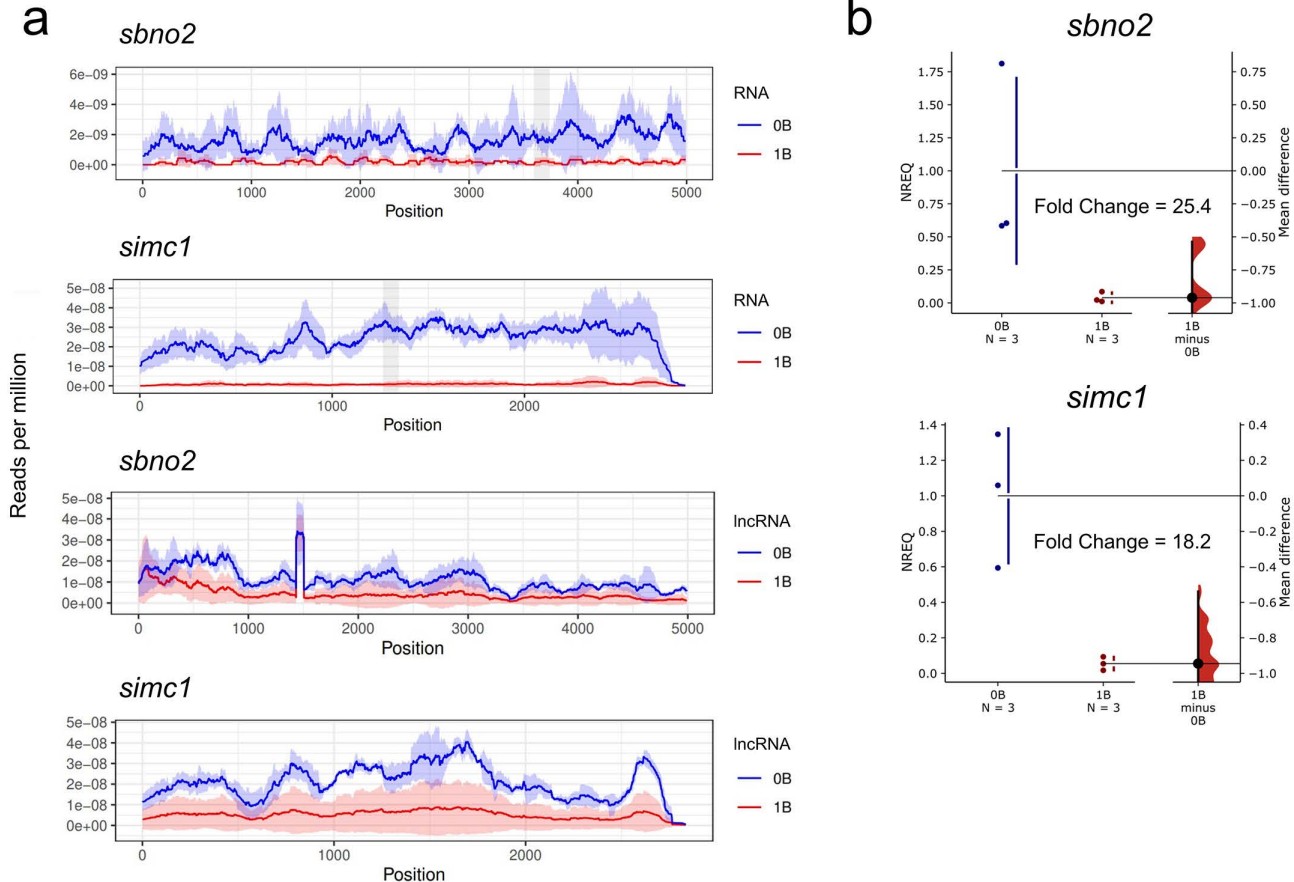

**Fig 2. Gene expression analysis of the pseudogenes on the B chromosomes of *Psalidodon* species. a** coverage graphs show underexpression of *Psalidodon* genes *sbno2* and *simc1* in samples with B chromosomes by RNAseq and lncRNAseq analysis. The shaded areas indicate the RT-qPCR amplicon. **b** Gardner-Altman estimation plots showing *sbno2* and *simc1* transcription levels in 0B and 1B individuals, analyzed by RT-qPCR. Both groups are plotted on the left axes, and the mean difference (effect size) is plotted on a floating axis on the right as a bootstrap sampling distribution. The mean difference is depicted as a black dot, and the 95% confidence interval is indicated by the ends of the vertical error bar. NREQ = normalized relative expression quantity.

with *P. fasciatus*. For the *sbno2-B*, the two *P. fasciatus* B chromosomes share more SNPs than with *P. paranae* and for *simc1-B*, the *P. paranae* and *P. bockmanni* B chromosomes share more SNPs than both with the *P. fasciatus* B chromosomes (Fig 1d, 8-S15 Tables in S1 Table). These findings support the idea that the B pseudogenes *sbno2-B* and *simc1-B* originated on an ancestral B chromosome, which persisted through speciation events in the genus, maintaining a relatively conserved structure over a long evolutionary period.

After the initial colonization of the B chromosome by the retrotransposed copies, the pseudogenic sequences likely underwent multiple rounds of amplification to achieve the higher copy numbers observed (Fig 1a). PacBio long-read analysis showed that the flanking regions of the *sbno2-B* belong to different parts of the B chromosome, indicating the *sbno2-B* was dispersed during amplification events, which was reinforced by the presence of FISH signals throughout the arms of the B chromosome. For the *simc1-B*, we found three sequences repeated in tandem in the PacBio long reads, which suggests that they have the potential to evolve into a satellite DNA structure. Also, for *simc1-B*, we observed just one group of unique sequences flanking of the pseudogene, indicating that the in-tandem sequences of this pseudogene are confined to a specific area of the B chromosome.

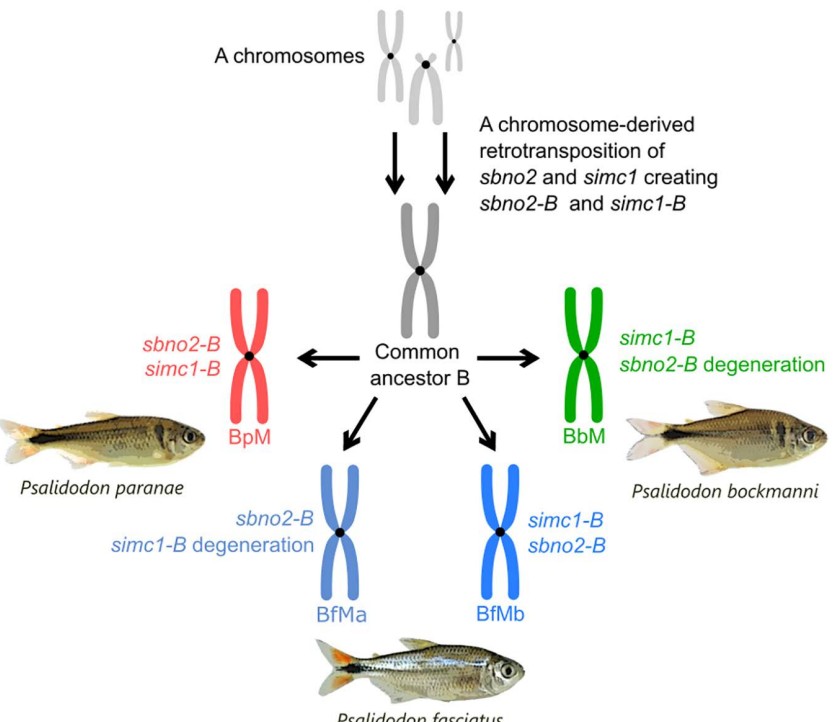

**Fig 3. Illustration of the most parsimonious hypothesis on how the B chromosomes of the *Psalidodon* species acquired and lost (in two cases) the pseudogenes *sbno-2-B* and *simc1-B*.**

The BbM and BfMa variants lack any copies of the *sbno2-B* and *simc1-B* pseudogenes, respectively. The most parsimonious hypothesis to explain this absence is that these sequences would have been lost during the evolutionary history of each B chromosome within each species. Degenerative processes such as random deletions, the formation of satellite DNAs or the insertion of other transposable elements may be responsible for these losses, as observed in sex chromosomes [33–35].Interestingly, a B chromosome variant of another *P. bockmanni* population has the *sbno2-B* pseudogene [13], suggesting that the loss of the *sbno2-B* pseudogene on the BbM variant analyzed here is a recent event.

The finding of reads in the exon-exon junctions 8–9 and 9–10 of the *sbno2* gene in 0B individuals of *P. paranae* indicates the presence of retrotransposed copies of this gene in the chromosomes of the standard A complement. Two hypotheses can explain this finding. The first is that, after retrotransposition of the *sbno2-B* copies onto the B chromosome from A chromosomes, a pseudogene fragment was moved back to the A chromosomes. The retrotransposition from the B to the A chromosomes could be facilitated by the presence of retrotransposon machinery surrounding the *sbno2-B* sequences on the B chromosome. This machinery would be a remnant of the initial retrotransposition process. Thus, both processes would be related. In all B chromosomes analyzed, the *sbno2-B* and *simc1-B* pseudogenes had high coverage across all exons, suggesting the entire pseudogenes were amplified. In contrast, on the A chromosomes, only two connected exon-exon junctions of *sbno2-B* were found, indicating that just part of the pseudogene was transferred back to the A chromosomes. The retrotransposition from the B to the A chromosomes could be facilitated by the presence of a retrotransposon machinery sequence surrounding downstream of the *sbno2-B* sequences on the B chromosome. This machinery sequence would be a remnant of the initial retrotransposition process.

The second hypothesis is that the retrotransposed A copies were created in an independent event of the B copies. This hypothesis is less likely because it relies on two independent retrotransposition capturing fragments of the same

gene. However, this could also be facilitated by retrotransposon sequence surrounding the canonical copy of the gene in the A chromosomes. Illumina 0B reads were found mapped in exon-exon junctions just in *P. paranae*, indicating that the retrotransposition of the *sbno2-B* pseudogene in 0B animals is exclusive of this species and that it is a later event compared to the retrotransposition to the B chromosome as the B pseudogene was retrotransposed in a common ancestor B chromosome.

In this study, we measured the expression levels of the canonical *sbno2* and *simc1* gene, both of which showed lower expression levels in the 1B samples. This differential expression may result from the regulation of canonical gene copies by the *sbno2-B* and *simc1-B* truncated pseudogenes. Truncated copies could be involved in canonical copy post-transcriptional direct or indirect regulation by mechanisms such as small siRNAs, miRNAs or competition for transcription factors [34,36] SNPs analysis using lncRNA reads did not show expression of B copies with B-specific variation. On the other hand, we observed some B-specific transcripts in mRNA reads for the *sbno2-B* pseudogene, indicating that this pseudogene can express mRNA truncated copies, which could be involved in canonical copy regulation by RNAi mechanisms. How these retrocopies are expressed in a new location probably lacking their parental regulatory elements, e.g., promoters and enhancers, is an intriguing question. However, retrocopies can benefit from regulatory elements of other genes in their vicinity [37]. We did not find B-specific SNPs in mRNA reads for the *simc1-B* and we therefore speculate that, if expressed, the *simc1-B* may produce other types of RNA, such as miRNAs.

If the *sbno2-B* and *simc1-B* pseudogenes can influence the expression of their canonical counterparts, we suspect that in the populations there is lacking one of the pseudogenes on the B chromosome, such as *P. bockmanni* (Segredo waterfall) and *P. fasciatus* (Alambari stream), the expression of the standard A-copies remains unaffected. This would lead to different effects of each B in the regulatory networks for each population. We did not measure the expression of these genes in the population besides the *P. paranae* one due to the difficulty of obtaining an adequate sample size for the RT-qPCR analysis. Therefore, identifying populations with favorable conditions to test these assumptions would be valuable for future research.

The underexpression of the *sbno2* and *simc1* genes in *P. paranae* 1B individuals may benefit B chromosome transmission across generations by manipulating the cell cycle. As the B chromosomes of *Psalidodon* species lack homologs to pair during meiosis, they employ an intriguing strategy of auto-pairing and gene expression manipulation to overcome the meiotic checkpoints [23]. Insights into how the *sbno2* and *simc1* genes might contribute to this process can be hypothesized based on the known roles of these genes in mammals. In mammals, the *Sbno2* gene is associated with cancer cell proliferation and survival [38], while the *Simc1* gene encoded protein, SIMC1, interacts with the CTBP1 protein, which is involved in the ability of cancer cells to evade cell-cycle checkpoints [39]. Collectively, we propose that the underexpression of both genes in the 1B samples may be crucial for the 1B cells to evade cell-cycle checkpoints and proliferate, ensuring B transmission over generations [24,10].

Finally, we propose a new protocol for the rapid identification of *Psalidodon* samples carrying large metacentric B chromosomes. Our results have shown that the HotShot DNA extraction protocol combined with multiplex PCR reactions to be an efficient tool for identifying B chromosomes in the three sampled *Psalidodon* species. With this tool the genotyping of large numbers of 0B and 1B animals is significantly expedited, reducing the processing time from a few days using classical cytogenetic analysis, to just a few hours. Furthermore, because fin clips are used for DNA extraction, this protocol allows for the minimally invasive genotyping of live specimens, thereby minimizing the impact on the studied populations and opening new avenues for various types of studies.

In conclusion, we demonstrated that the metacentric B chromosomes of three *Psalidodon* species share two retrotransposed pseudogenes, *sbno2-B* and *simc1-B*, which were acquired from an ancestral B chromosome. Canonical copies of the *sbno2* and *simc1* genes are underexpressed in 1B animals in *P. paranae*, indicating that pseudogenic B copies influence their expression through non-coding RNA interference mechanisms. Fragments of the *sbno2-B* pseudogene were identified on the standard A chromosomes, showing that the dispersion of these genes contributes to genomic diversity in

both the A and B chromosomes. We did not precisely determine whether these A-copies of the pseudogenes are present in one or more A chromosomes, or which specific chromosomes they might occupy. Future work, including whole-genome assemblies of *Psalidodon* species and in-depth analysis of transposable elements on the B chromosomes, will be crucial in better understanding how these pseudogenes are organized on both the A and B chromosomes. Our work would benefit from a dendrogram including the *sbno2* and *simc1* paralogs to investigate the origins of the pseudogenes on the B and A chromosomes. However, the sequences did not exhibit sufficient variation for this analysis.

## Supporting information

**S1 Fig. Illustration of the primer scheme for multiplex and regular PCR to confirm the presence of B chromosome-specific sequences on samples with B chromosomes.** F1 = forward primer. R1 = reverse primer 1. R2 = reverse primer 2.
(PDF)

**S2 Fig. Summary of the investigation approach.**
(PDF)

**S3 Fig. Coverage graph of the *sbno2* gene on the B chromosomes of *Psalidodon paranae* excluding the high-coverage region to visualize the absence of differences between 0B and 1B samples.**
(PDF)

**S4 Fig. Coverage graph of the full *sbno2* (a, c) and *simc1* (b, d) genes sequences of *Psalidodon paranae* (including introns).** The high-coverage peaks observed in c, d were removed from the analysis in a, b for better visualization of the low-coverage regions. The shaded areas indicate the exon boundaries.
(PDF)

**S5 Fig. Example of *Psalidodon paranae* PacBio reads alignment with the pseudogene sequences annotated, *sbno2* (green) (a) and *simc1* (pink) (b).**
(PDF)

**S6 Fig. Example of Illumina short reads mapped to a reference sequence.** The dotted lines show the exon-exon junctions. Note the presence of reads crossing the exon-exon junctions in all the mappings. In b, the red boxes identify the reads crossing the exon-exon junctions for visualization purposes.
(PDF)

**S7 Fig. Coverage graph using 0B reads of two *P. paranae* individuals.** The dotted lines show the exon-exon junctions. The sample highlighted in blue showed reads mapped in the exon-exon junctions. Note the slightly higher coverage in the blue sample on exons present in the 0B genome (green boxes) compared to its flanks (exons 7 and 11) and the red sample.
(PDF)

**S1 Raw images. Uncropped raw images of the electrophoresis gels used in this study.**
(PDF)

**S1 File. TAREAN clustering results for the *sbno2* reads of *Psalidodon paranae*.**
(PDF)

**S2 File. TAREAN clustering results for the *simc1* reads of *Psalidodon paranae*.**
(PDF)

**S1 Table. Supplementary tables for this study.**
(XLSX)

## Acknowledgments

The authors thank Monica Renee Stein for the English editing.

## Author contributions

**Conceptualization:** Ricardo Utsunomia, Duilio Mazzoni Zerbinato A Silva, Fausto Foresti.

**Data curation:** Duilio Mazzoni Zerbinato A Silva.

**Formal analysis:** Lucas F. Lasmar, Ricardo Utsunomia, Duilio Mazzoni Zerbinato A Silva.

**Funding acquisition:** Duilio Mazzoni Zerbinato A Silva, Fausto Foresti.

**Investigation:** Lucas F. Lasmar, Mateus R. Vidal, Pamela C. F. Nadai, Rodrigo Zeni dos Santos, Raquel da Costa Machado, Ricardo Utsunomia, Duilio Mazzoni Zerbinato A Silva.

**Methodology:** Rodrigo Zeni dos Santos, Francisco J. Ruiz-Ruano, Alexander Suh, Duilio Mazzoni Zerbinato A Silva, Fausto Foresti.

**Project administration:** Claudio Oliveira, Duilio Mazzoni Zerbinato A Silva, Fausto Foresti.

**Resources:** Claudio Oliveira, Fausto Foresti.

**Software:** Francisco J. Ruiz-Ruano.

**Supervision:** Francisco J. Ruiz-Ruano, Alexander Suh, Claudio Oliveira, Duilio Mazzoni Zerbinato A Silva, Fausto Foresti.

**Validation:** Lucas F. Lasmar, Mateus R. Vidal, Pamela C. F. Nadai, Rodrigo Zeni dos Santos, Raquel da Costa Machado, Alexander Suh, Duilio Mazzoni Zerbinato A Silva.

**Visualization:** Lucas F. Lasmar, Mateus R. Vidal, Rodrigo Zeni dos Santos, Raquel da Costa Machado, Francisco J. Ruiz-Ruano, Alexander Suh, Duilio Mazzoni Zerbinato A Silva, Fausto Foresti.

**Writing – original draft:** Lucas F. Lasmar, Mateus R. Vidal, Pamela C. F. Nadai, Raquel da Costa Machado, Ricardo Utsunomia, Francisco J. Ruiz-Ruano, Claudio Oliveira, Duilio Mazzoni Zerbinato A Silva, Fausto Foresti.

**Writing – review & editing:** Lucas F. Lasmar, Mateus R. Vidal, Pamela C. F. Nadai, Rodrigo Zeni dos Santos, Ricardo Utsunomia, Francisco J. Ruiz-Ruano, Alexander Suh, Claudio Oliveira, Duilio Mazzoni Zerbinato A Silva, Fausto Foresti.

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
