## [Decision Letter · Decision Letter 0]

16 Oct 2025

Dear Dr. Mazzoni Zerbinato A Silva,

Thank you for submitting your manuscript to PLOS ONE. After careful consideration, we feel that it has merit but does not fully meet PLOS ONE’s publication criteria as it currently stands. Therefore, we invite you to submit a revised version of the manuscript that addresses the points raised during the review process.

We look forward to receiving your revised manuscript.

Kind regards,

James Lee Crainey, Ph.D.

Academic Editor

PLOS ONE

Journal Requirements:

3. Thank you for uploading your study's underlying data set. Unfortunately, the repository you have noted in your Data Availability statement does not qualify as an acceptable data repository according to PLOS's standards.

Additional Editor Comments:

Please attend to the very minor concerns raised by reviewer 1 and return your manuscript at the earliest opportunity.

Reviewer's Responses to Questions

**Comments to the Author**

1. Is the manuscript technically sound, and do the data support the conclusions?

Reviewer #1: Yes

Reviewer #2: Yes

2. Has the statistical analysis been performed appropriately and rigorously?

Reviewer #1: I Don't Know

Reviewer #2: Yes

3. Have the authors made all data underlying the findings in their manuscript fully available?

Reviewer #1: Yes

Reviewer #2: Yes

4. Is the manuscript presented in an intelligible fashion and written in standard English?

Reviewer #1: Yes

Reviewer #2: Yes

Reviewer #1: The authors demonstrated that the four B variants of three Psalidodon species contain retrotransposed copies of the sbno2 and simc1 genes. These gene copies were acquired in a common ancestor and have been maintained across speciation events. The findings are supported by bioinformatic analysis of gDNA and RNA reads, PCR, and FISH. The absence of introns in both genes suggests that retrotransposition caused the transfer of these gene fragments. Interestingly, it appears that both pseudogenes may influence the expression of their counterparts located on the A chromosomes. The quality of the results is high, and the findings are logically discussed in the context of current knowledge.

Line 45

Please consider clarifying that B chromosome drive mechanisms can occur pre-meiotically and post-meiotically, not just during meiosis, depending on the species.

Line 75 and others (e.g. line 201, )

Please avoid using the term “A genome” and instead use “A chromosomes.” The term “A genome” is typically reserved for distinguishing subgenomes in allopolyploid species, which could cause confusion.

Line 301

Please indicate where in the manuscript the quick protocol for identifying B-chromosome-bearing individuals is shown.

Figure 3.

Please add in the figure “A chromosome-derived retrotransposition of….”

Reviewer #2: Manuscript ID: PONE-D-25-45744

Title: B chromosome retrotransposed sequences persist through speciation, contributing to genomic and regulatory innovations in the fish genus Psalidodon (Characiformes, Acestrorhamphidae)

Authors: Lucas F. Lasmar, Mateus R. Vidal, Pamela C. F. Nadai, Rodrigo Zeni dos Santos, Raquel da Costa Machado, Ricardo Utsunomia, Francisco J. Ruiz-Ruano, Alexander Suh, Claudio Oliveira, Duilio M. Z. A. Silva, Fausto Foresti

The goal of the presente paper was to investigate the evolutionary history of two pseudogenes, sbno2-B and simc1-B, on the B chromosomes in Psalidodon species, as well as their potential effects on gene expression. The authors demonstrate that four B chromosome variants of three Psalidodon species have retrotransposed copies of the sbno2 and simc1 genes that were acquired in a common ancestor and maintained across speciation events.

B chromosomes and retrotransposons are distinct genetic elements that share a remarkable ability to selfishly take advantage of biological processes, thereby increasing their transmission to future generations through different molecular mechanisms. B chromosomes are supernumerary genetic elements that primarily cheat by preferentially being incorporated into generative cells during meiosis. In contrast, retrotransposons are genomic sequences that over-replicate during cell cycles.

The work is of great interest, primarily because it addresses two distinct and intriguing genetic elements. In this study, the authors demonstrated that retrotransposons have activity on B chromosomes, generating genomic diversity. Furthermore, Psalidodon spp. can be considered “model species” for the study of B chromosomes.

I recommend your publication.

One correction is pointed out:

Page 2, Line 58: “The genus Psalidodon Eigenmann, 1911 within the family Characiformes includes about 30 species,…”

within the Family Acestrorhamphidae or order Characiformes

**Do you want your identity to be public for this peer review?** For information about this choice, including consent withdrawal, please see our Privacy Policy

Reviewer #1: No

Reviewer #2: No

---

## [Author Response · Author response to Decision Letter 1]

30 Nov 2025

Dear James Lee Crainey,

Academic Editor

We sincerely appreciate the time and effort you and the reviewers have invested in evaluating our manuscript for PLOS ONE. We are pleased with the overall positive feedback and have carefully addressed the suggestions provided. We have used track changes to highlight the modifications made in response to the comments.

Thank you for considering our revised manuscript.

Cordially,

Duílio Mazzoni Zerbinato de Andrade Silva

Editor

1.Please ensure that your manuscript meets PLOS ONE's style requirements, including those for file naming. The PLOS ONE style templates can be found at https://journals.plos.org/plosone/s/file?id=wjVg/PLOSOne_formatting_sample_main_body.pdf and https://journals.plos.org/plosone/s/file?id=ba62/PLOSOne_formatting_sample_title_authors_affiliations.pdf

We conducted a thorough review of the entire manuscript to guarantee it meets PLOS ONE’S style requirements.

We reviewed all the manuscript figures to ensure they adhere fully to the PLOS ONE guidelines. Also, we provided all the original gel images as Supporting Information.

3. Thank you for uploading your study's underlying data set. Unfortunately, the repository you have noted in your Data Availability statement does not qualify as an acceptable data repository according to PLOS's standards.

We provided the GenBank accession codes for the deposited sequences.

The reviewers’ suggestions did not include recommendations to cite specific previously published works. Thus, we addressed and followed all the reviewers’ suggestions to the manuscript.

We conducted a thorough review of the reference list to ensure it is complete and correct.

Reviewer #1

Line 45

Please consider clarifying that B chromosome drive mechanisms can occur pre-meiotically and post-meiotically, not just during meiosis, depending on the species.

We thank the reviewer for the suggestion. We clarified in lines 45-46 that B chromosome accumulation mechanisms can also occur at pre-meiotic and post-meiotic stages.

Line 75 and others (e.g. line 201, )

Please avoid using the term “A genome” and instead use “A chromosomes.” The term “A genome” is typically reserved for distinguishing subgenomes in allopolyploid species, which could cause confusion.

To avoid confusion with the terminology used for allopolyploid species, we have replaced all the occurrences of “A genome” with “A chromosomes” throughout the manuscript.

Line 301

Please indicate where in the manuscript the quick protocol for identifying B-chromosome-bearing individuals is shown.

We have now clarified in the text where the quick protocol is described.

Reviewer #2

Page 2, Line 58: “The genus Psalidodon Eigenmann, 1911 within the family Characiformes includes about 30 species,…” within the Family Acestrorhamphidae or order Characiformes

We thank the reviewer for the suggestion. We have corrected the sentence accordingly to refer to the order Characiformes.

---

## [Editor Report · Decision Letter 1]

16 Dec 2025

B chromosome retrotransposed sequences persist through speciation, contributing to genomic and regulatory innovations in the fish genus Psalidodon (Characiformes, Acestrorhamphidae)

PONE-D-25-45744R1

Dear Dr. Mazzoni Zerbinato A Silva,

We’re pleased to inform you that your manuscript has been judged scientifically suitable for publication and will be formally accepted for publication once it meets all outstanding technical requirements.

Kind regards,

James Lee Crainey, Ph.D.

Academic Editor

PLOS One
---

## [Editor Report · Acceptance letter]

PONE-D-25-45744R1

PLOS One

Dear Dr. Mazzoni Zerbinato A Silva,

I'm pleased to inform you that your manuscript has been deemed suitable for publication in PLOS One. Congratulations! Your manuscript is now being handed over to our production team.

Kind regards,

on behalf of

Dr. James Lee Crainey

Academic Editor

PLOS One